# Negative effects of agricultural open-channel irrigation system on vertebrate populations in central Mexico

Yuriana Gómez-Ortiz[1], Hublester Domínguez-Vega[1], Leroy Soria-Díaz[2], Tamara Rubio-Blanco[1], Claudia C. Astudillo-Sánchez[3], Victor Mundo[4] and Armando Sunny[5]

[1] División de Desarrollo Sustentable, Universidad Intercultural del Estado de México, San Felipe del Progreso, México, Mexico
[2] Instituto de Ecología Aplicada, Universidad Autónoma de Tamaulipas, Ciudad Victoria, Tamaulipas, Mexico
[3] Facultad de Ingeniería y Ciencias, Universidad Autónoma de Tamaulipas, Ciudad Victoria, Tamaulipas, Mexico
[4] Campus Universitario Siglo XXI, Grupo Educativo Siglo XXI, Toluca, México, Mexico
[5] Centro de Investigación en Ciencias Biológicas Aplicadas, Universidad Autónoma del Estado de México, Toluca, México, Mexico

## ABSTRACT

Linear infrastructures such as agricultural irrigation channels produce physical changes and negative impacts to habitats, wildlife populations, communities, and ecosystems. Open irrigation channels act as a pitfall for wildlife and can affect vertebrates of all sizes. Nonetheless, small channels have received relatively little attention by conservation biologists. The objective of this study was to analyze vertebrate species richness and mortality in relation to different sections of an irrigation channel system and the surrounding landscape characteristics. For two years, we conducted monthly surveys along an open-channel irrigation system to estimate its effect on vertebrates through records of dead and alive individuals. We examined the spatial relation of species richness and mortality with transects using a canonical correspondence analysis and chi-squared tests to determine possible variations in the different structures of the channel and seasonality. Further, a landscape diversity index was used to analyze the importance of surrounding habitat structure and composition on these parameters. Most vertebrates (61%) were found dead, small mammals and reptiles were the most affected. Our results indicate that mortality of small vertebrates varies depending on species, structures of the open-channel agricultural irrigation system (*i.e.*, concrete channel and floodgates), seasonality (*i.e.*, wet, and dry), and landscape heterogeneity (*i.e.*, high, medium, and low landscape diversity). The open-channel irrigation system is a threat to populations of small vertebrates in anthropized landscapes, conservation efforts should be directed at protecting water bodies and restructuring the open-channel agricultural irrigation system to avoid mortality of species such as small rodents (*M. mexicanus*) and reptiles (*C. triseriatus*, *B. imbricata*, and *Thamnophis* spp.).

Corresponding author
Hublester Domínguez-Vega, hublester.dvega@gmail.com

## INTRODUCTION

Human-induced landscape changes are the main cause of habitat fragmentation, degradation, and loss (*Fahrig, 2003*; *van der Ree et al., 2011*). Agriculture is one of the most conspicuous human-made landscapes around the world and is also considered a global threat to biodiversity conservation (*Canelas & Pereira, 2022*; *Foley et al., 2011*; *Lemly, Kingsford & Thompson, 2000*; *Munstermann et al., 2021*). Open channel irrigation systems constitute the functional core of agricultural landscapes, where approximately 70% of the water consumed globally is used (*Coronel-Arellano et al., 2021*). Linear infrastructures such as agricultural irrigation channels produce physical changes and negative impacts on habitats, wildlife populations, communities, and ecosystems (*van der Ree et al., 2011*).

The impact of irrigation channels (and other linear infrastructures), includes two main types of disturbance to wildlife; the most visible is the direct impact through mortality of individuals. Some other studies evidence the indirect effects through isolation, as they function as barriers to movement of some species (*Ascensão et al., 2019*; *Beebee, 2013*; *Cushman, 2006*; *García, 2009*). In particular, irrigation channels function as pitfalls where animals are accidentally trapped during their daily movement. In addition, depending on their dimensions, irrigation channels can affect vertebrates of all sizes, from small species as amphibians to big mammals including livestock (*Arranz, 1994*; *García, 2009*; *Peris & Morales, 2004*). The impact of irrigation channels on wildlife depends on variables such as the design, the materials used for its construction and the composition of vegetation on the channel shores and in the surrounding area (*López-Polomares, López-Iborra & Martín-Cantarino, 2015*).

Several studies have shown that linear infrastructures (*i.e.,* water channels, roads, railways, power lines, pipelines, wind farms and fences), can increase mortality rates and cause wildlife populations to decline significantly and even become locally extinct (*Ascensão et al., 2019*; *Jones, 2000*; *Sergio et al., 2004*; *Shaw et al., 2010*). Although irrigation channels are also considered a major cause of direct wildlife mortality, they have received relatively little attention. Nonetheless, some empirical evidence shows that several groups may be strongly affected. For example, *Peris & Morales (2004)*, during a study of 5 years, recorded 538 dead mammals (including domestic and wild species), whilst *García (2009)* found that amphibians are the most affected biological group of vertebrates (86.4%) in a water channel in Spain.

Irrigation channels are considered priority infrastructures in an increasingly water-thirsty world. The number of irrigation channels is on the rise in almost all regions and the existing literature proves that they deserve much more attention than they have received from conservation biologists. In addition, most literature on this subject is focused on relatively big channels, where large mammals frequently drown (*e.g.*, *Odocoileus hemionus*, *Pecari tajacu*, *Sus scrofa*, *Mazama gouazoubira*, *Tapirus terrestris*; *Albanesi, Jayat & Brown, 2016*; *Bucci & Krausman, 2015*; *Peris & Morales, 2004*). Thus, impacts on small vertebrates are likely to be underestimated due to most studies are focused on big channels where small species may be either undetected or unaffected, also due to small channels has been scarcely studied.

We aimed to analyze three aspects related to small vertebrates mortality in an open channel irrigation system to understand its direct impacts on small vertebrates and how the surrounding landscape may change such effects: (1) the effect of structures composing the irrigation channel (open channel and floodgates), (2) mortality changes related to seasonality, and (3) the effect of surrounding landscape characteristics.

# MATERIALS AND METHODS

## Study area

The study area is in the peripheral zone of the urban area of Toluca, Mexico. We monitored an open-channel irrigation system located between an agricultural zone and the protected area "Area for the conservation and ecological research, Bordo Las Maravillas" (99.68°W, 19.4°N, 2,614 masl). The agricultural open-channel irrigation system consists of a linear infrastructure including the irrigation channel and several floodgates along. The complete irrigation system is 1,470 m long, it is divided into three linear transects and surrounded by agricultural fields (Fig. 1). Additionally, there are two water bodies near the channel; both are surrounded by native grasses (*Muhlenbergia* sp. and *Festuca* sp.) and aquatic vegetation (*Salix babylonica*, *Scirpus* sp. and *Typha latifolia*). The irrigation channel is a relatively small and open structure made of concrete (0.40 m low base, 0.77 m high base and 0.50 m height). It has 14 floodgates distributed along (mean distance between floodgates = 184.41m) with depth ranging from 0.67 to 2 m and open area from 0.69 $m^2$ to 1.18 $m^2$. The open-channel agricultural irrigation system carries water mostly in the rainy season, between May and September, while the rest of the year remains dry. This open-channel irrigation system can be considered as a functional unit of the agricultural system in the region (although they may present great variability in their structural and management characteristics), which includes more than 1,500 $km^2$, where more than 60% of the land is devoted to agriculture. In this region there are more than 1,000 water bodies that are used as reservoirs to distribute water to neighboring farms using similar systems (*INEGI, 2015*).

## Surveys of wildlife and habitat features

We conducted monthly surveys along four transects for approximately two years (June 2014–November 2016), searching for small vertebrates in three transects along the irrigation channel including the floodgates, and one transect along the native vegetation in the protected area that provides the water for the irrigation channel. In each survey, at least four people performed the search on foot using the VES (Visual Encounter Survey), technique (*McDiarmid et al., 2012*). We used globes to manipulate dead individuals, and non-venomous live individuals; for the live venomous snakes (*C. triseriatus*), we used herpetological hooks to relocate individuals. The number of dead and alive vertebrates in the four transects was recorded. All dead individuals were removed from the channel to avoid double counting in the next surveys, and the live individuals were relocated on native vegetation outside the channel. We recorded the name of the species and assigned its location to the corresponding transect. The records were also classified according to its location within the irrigation channel as "inside" or "outside" the irrigation channel, and when in floodgates.

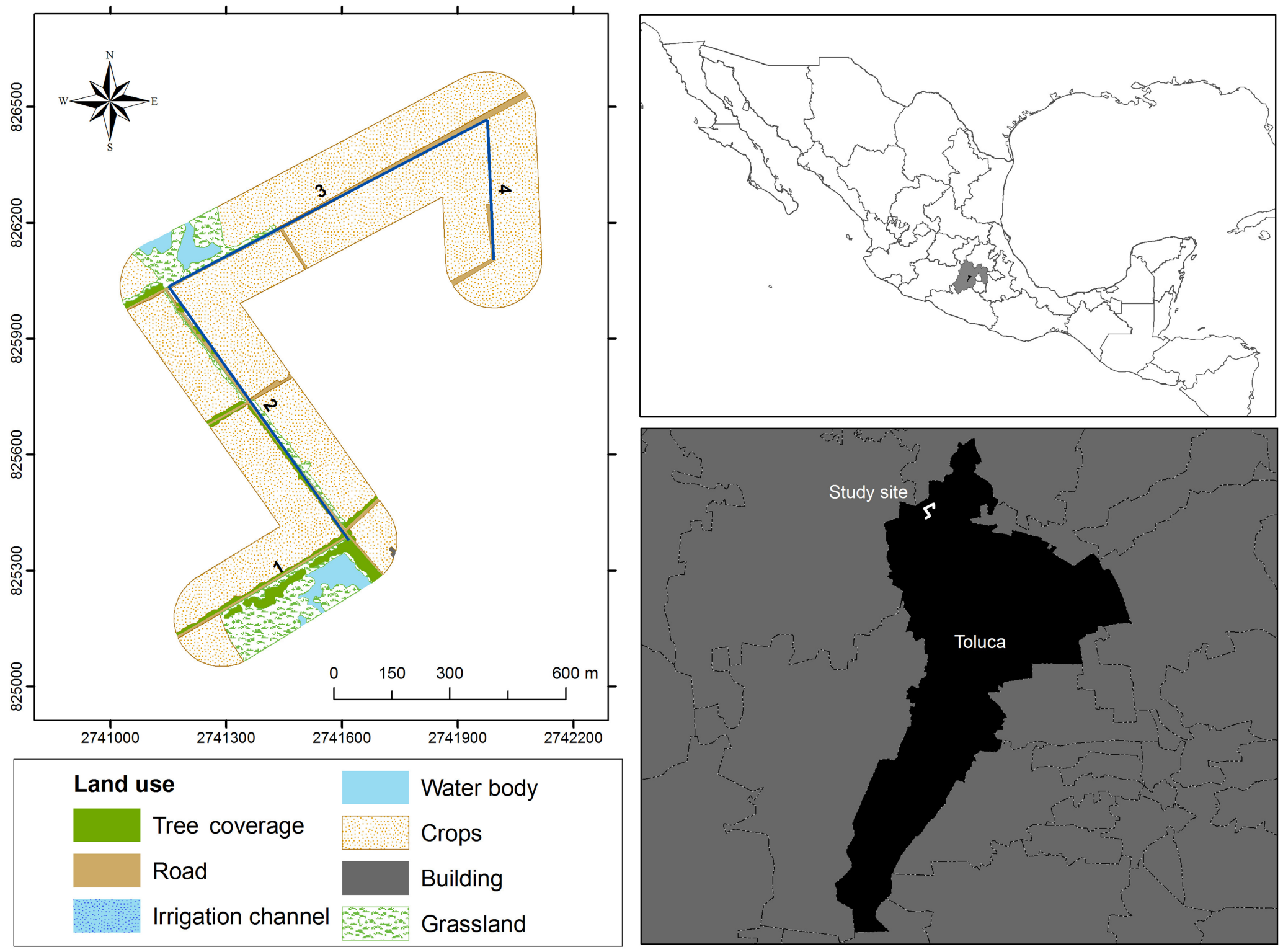

**Figure 1 Location of the study area in the peripheral zone of the urban area in Toluca, Mexico.** The open channel irrigation system includes four transects along an agricultural zone and the protected area "Area for the conservation and ecological research, Bordo Las Maravillas", Toluca, Mexico State. The studied transects are numbered from 1 to 4.

In addition, we analyzed the relationship of the surrounding landscape with species richness and the frequency of occurrence of dead and alive individuals using a landscape diversity index. To this aim, a 250 m buffer along each transect surveyed in the irrigation system was created and then we classified land cover within the buffers based on a 0.4 m resolution satellite image using ArcGIS 10.2. The land cover classification was verified by ground truthing (*i.e.,* surveying the ground to ensure that all landscape features matched those on the image). We identified four land cover classes (*i.e.*, grassland, crops, trees and water bodies). The landscape diversity index was calculated for the surrounding landscape of each transect using Shannon's diversity index (available in Patch Analyst Extension; *Elkie, Rempel & Carr, 1999*). To estimate the landscape diversity index, the number of land cover classes per transect are used, considering the total number of patches and the size of those patches (in squared meters), in each transect. The landscape diversity index is homologous to the diversity index frequently used in biodiversity analysis. In our study, indicates the dominance of one land cover class over the others (*i.e.*, low values correspond to a homogeneous landscape context, while high values correspond to a heterogeneous landscape context). In our study area, homogeneous landscape contexts are associated with crop dominance, while heterogeneous contexts include water bodies and native grasslands.

The Secretaría del Medio Ambiente y Recursos Naturales (SEMARNAT), provided permissions for this work through the number (FAUT-0351).

### Statistical analysis

Firstly, we used a descriptive approach to analyze the frequency of occurrence and death rate of each species in the channel system. Then we used Chi-square tests to compare the frequency of mortality between individuals' location (channel *vs* floodgates) and between seasons (rainy season from May to September and dry season from October to April). In this regard, we applied a fourfold contingency table (*Zar, 2010*), to test the null hypothesis of no differences in the ratios of dead individuals between the channel irrigation system location (open channel *vs* floodgates), and no differences in the ratios of dead individuals between seasons (dry *vs* wet). For each comparison, we used the total number of dead and live individuals assigned to its location or season, considering each record of dead or alive individual as an independent event. We used the Chi-squared test because it allows the analysis of two nominal scale variables (in our case, dead or alive individuals *vs* location or season), and it has been used to comparing resource use by wildlife (*Thomas & Taylor, 1990*; *Jelinski, 1991*). In order to analyze the importance of landscape in the channel effects, we explored the spatial association between species and transects using a canonical correspondence analysis. This ordination analysis shows the relation of each species with transects using the frequency of records in those transects and showing in which of the transects each species is more commonly found. Finally, we used the landscape diversity index along with the canonical correspondence analysis to discuss the relationship between species richness and mortality rate with landscape heterogeneity.

## RESULTS

We recorded 227 individuals of 10 species from four vertebrate groups: Birds (one species), Amphibians (two species), Mammals (three species) and Reptiles (four species); 61% of the records (139) were dead individuals belonging to eight species (we recorded no dead individuals of *Dryophytes eximius* and *Sceloporus* spp.). Reptiles (*Crotalus triseriatus*, *Barisia imbricata* and *Thamnophis spp.*) and small mammals (*Cryptotis parva*, *Microtus mexicanus* and *Neotoma* sp.) accounted for most dead individuals (51.1% and 46.8%, of all records respectively), while birds (*Anas diazi*) and amphibians (*Ambystoma granulosum*) together accounted for only 2.2% of dead individuals (Fig. 2).

### Occurrence and mortality between channel and floodgates

Most dead individuals (99%) were found in the agricultural open-channel irrigation system, both along the channel and in the floodgates. The frequency of dead and alive individuals is significantly different between the channel and the floodgates ($X^2 = 11.07$; g.l. =1; $p < 0.001$); we found more dead individuals in the floodgates (58.40%) than in the channel (41.60%; Fig. 3). The Mexican vole (*M. mexicanus*) had the highest percentage of mortality in the floodgates (73.75% of all dead individuals recorded in this structure), whilst for the channel, the Central Plateau rattlesnake (*C. triseriatus*; 54.39%), followed by the transvolcanic alligator lizard (*B. imbricata*; 31.58%) were the most affected species,

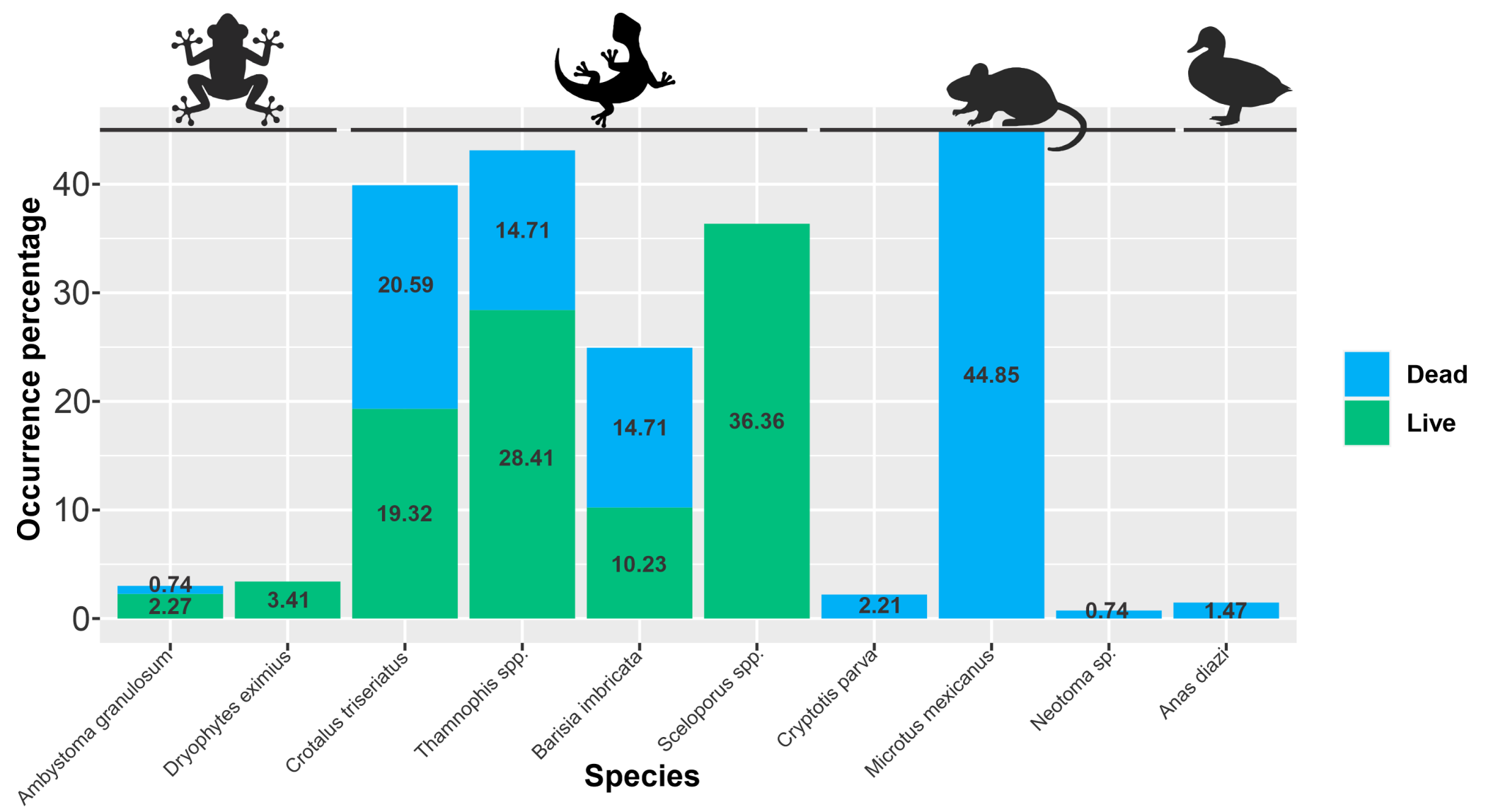

**Figure 2 Dead and alive small vertebrates found in an irrigation channel system.** Occurrence percentage of dead (blue), and alive (green) individuals per species found in the irrigation channel system. Silhouette image source credit: mouse, craft starters; frog, siridhata; Curved Lizard, designed by Freepik.

although the effect of predators and scavengers may be affecting these estimation, mainly in the concrete channel. Otherwise, we recorded more live individuals in the channel (67.22% of all vertebrates recorded in the irrigation system) than in the floodgates (32.78%). The species with the highest number of living individuals observed in the channel was *Sceloporus* spp. (78.04%), while the garter snakes *Thamnophis* spp. (70%) presented the highest frequency of alive individuals in the floodgates.

## Mortality differences between dry and wet seasons

Mortality of small vertebrates was significantly higher in the wet season (56.12%) than in the dry season (43.88%) ($X^2 = 7.55$; g. $l = 1$; $p < 0.05$; Fig. 4). In the wet season the Central Plateau rattlesnake (*C. triseriatus*) was the most affected species (39.74% of all dead vertebrates recorded in this season), followed by the Mexican vole (*M. mexicanus*, 23.07%) and the transvolcanic alligator lizard (*B. imbricata*, 19.23%), whilst in the dry season, the Mexican vole (*M. mexicanus*) was, by far, the most affected species (70.49% of all dead vertebrates recorded in this season). Granular salamanders (*A. granulosum*), small-eared shrews (*C. parva*) and woodrats (*Neotoma* sp.) were scarcely affected by drowning; and for *Dryophytes eximius* and *Sceloporus* spp., no dead individuals were recorded, suggesting that these species are not negatively affected by the irrigation channel. In this regard, predators and scavengers may be affecting our estimates for some species, particularly for those species commonly found in the concrete channel.

## Mortality differences between habitat features

The results of the land use diversity index showed that the four transects present differences in landscape context. Some transects are heterogeneous (transects 1 and

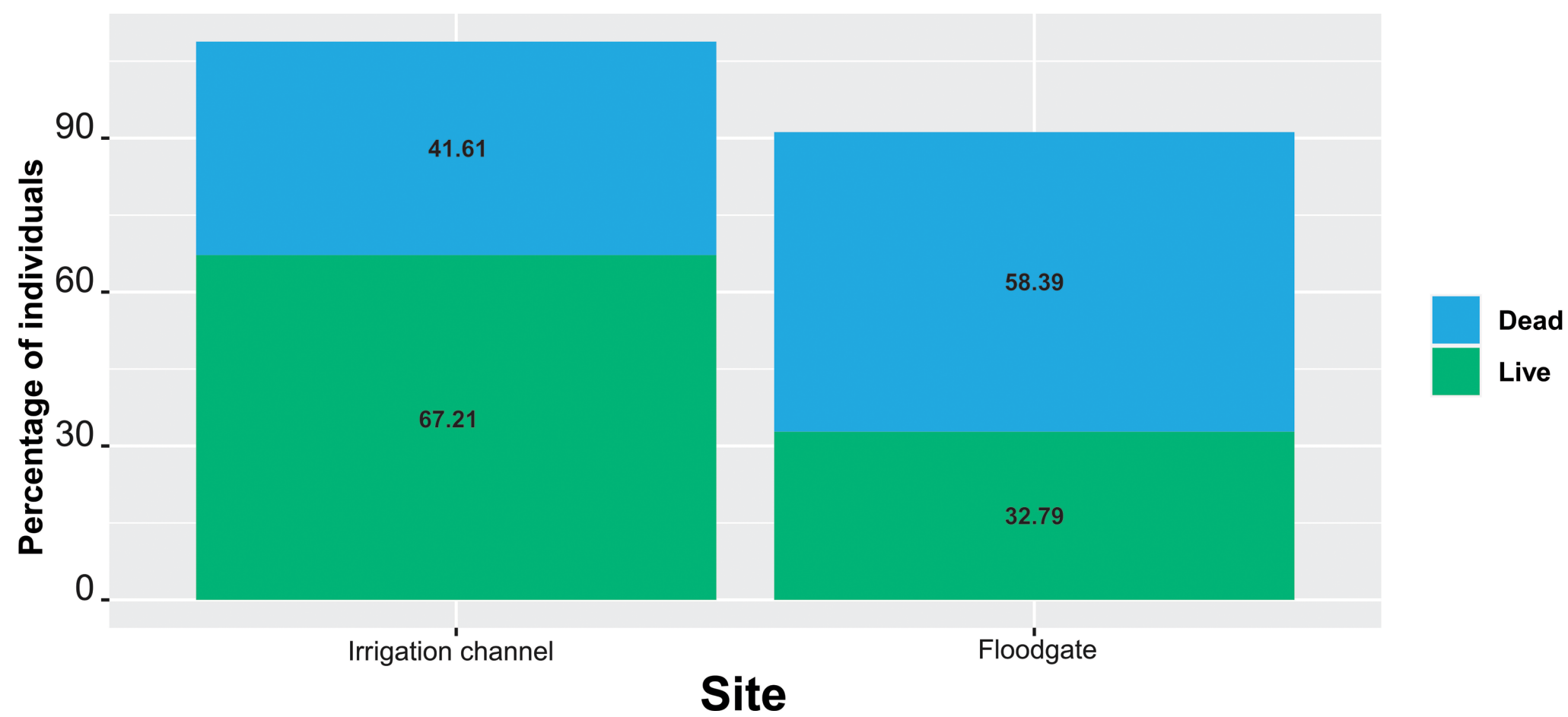

**Figure 3 Spatial distribution of alive and dead individuals of small vertebrates inside the irrigation channel and floodgates.** The percentage of each category (dead or alive) is shown. Note that the greater percentage of alive individuals were recorded in the irrigation channel whilst the greater percentage of dead individuals occurred in the floodgates. According with the chi-squared test, there are significant differences ( $X^2 = 11.07$; $g.l. = 1$; $p = 0.001$) in the mortality frequency associated with these components of the irrigation system.

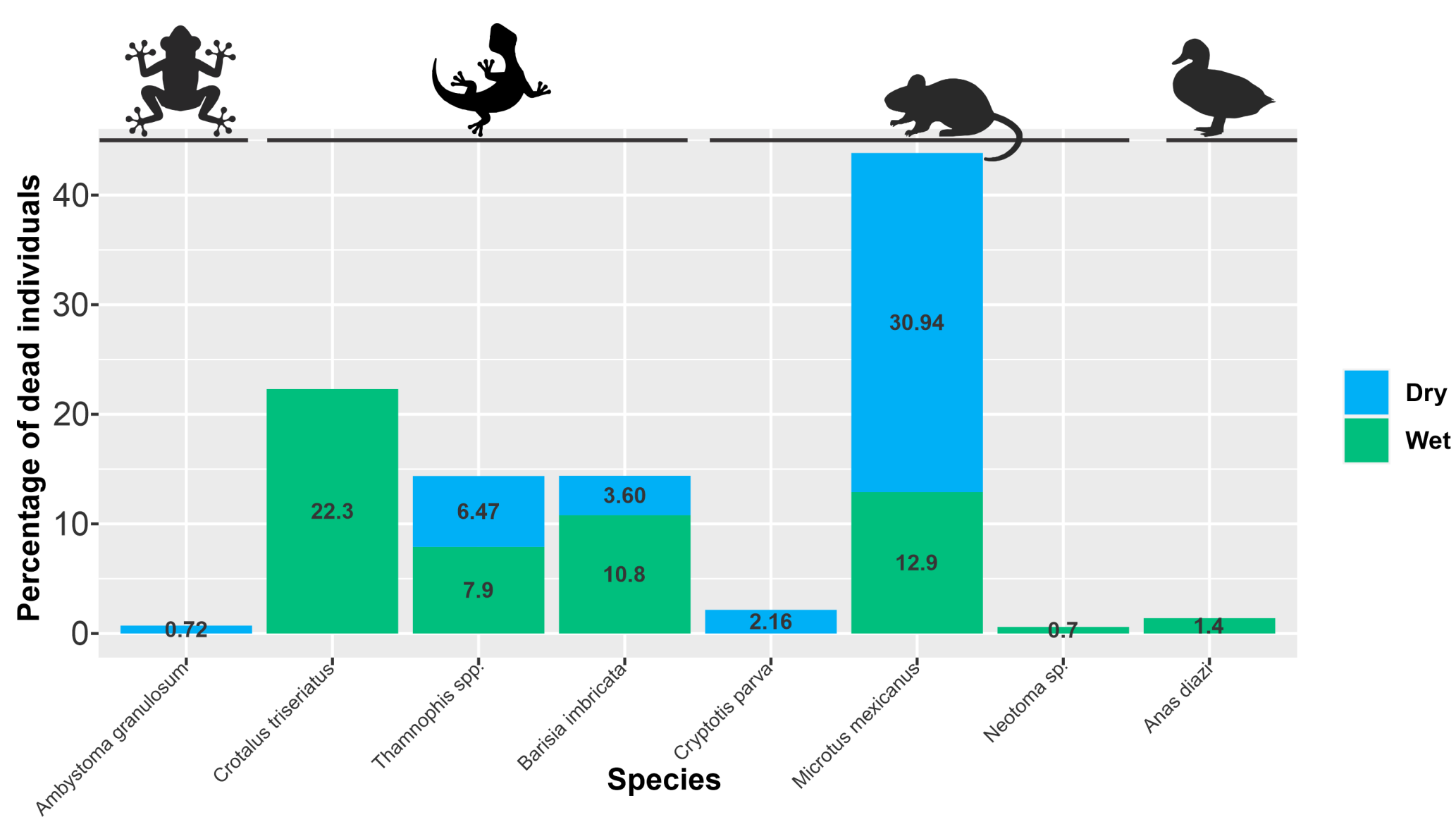

**Figure 4 Dead individuals recorded per season in the irrigation channel system.** Percentage of dead individuals recorded in the dry (blue), and wet (green) seasons. Silhouette image source credit: mouse, craft starters; frog, siridhata; Curved Lizard, designed by Freepik.

3), whilst others may be considered homogeneous (transects 2 and 4) in regards its land cover. Correspondence analysis and occurrence data show that most species are strongly associated with specific transects in the open-channel irrigation system, suggesting that landscape context may influence species occurrence and mortality. For example, *Sceloporus*

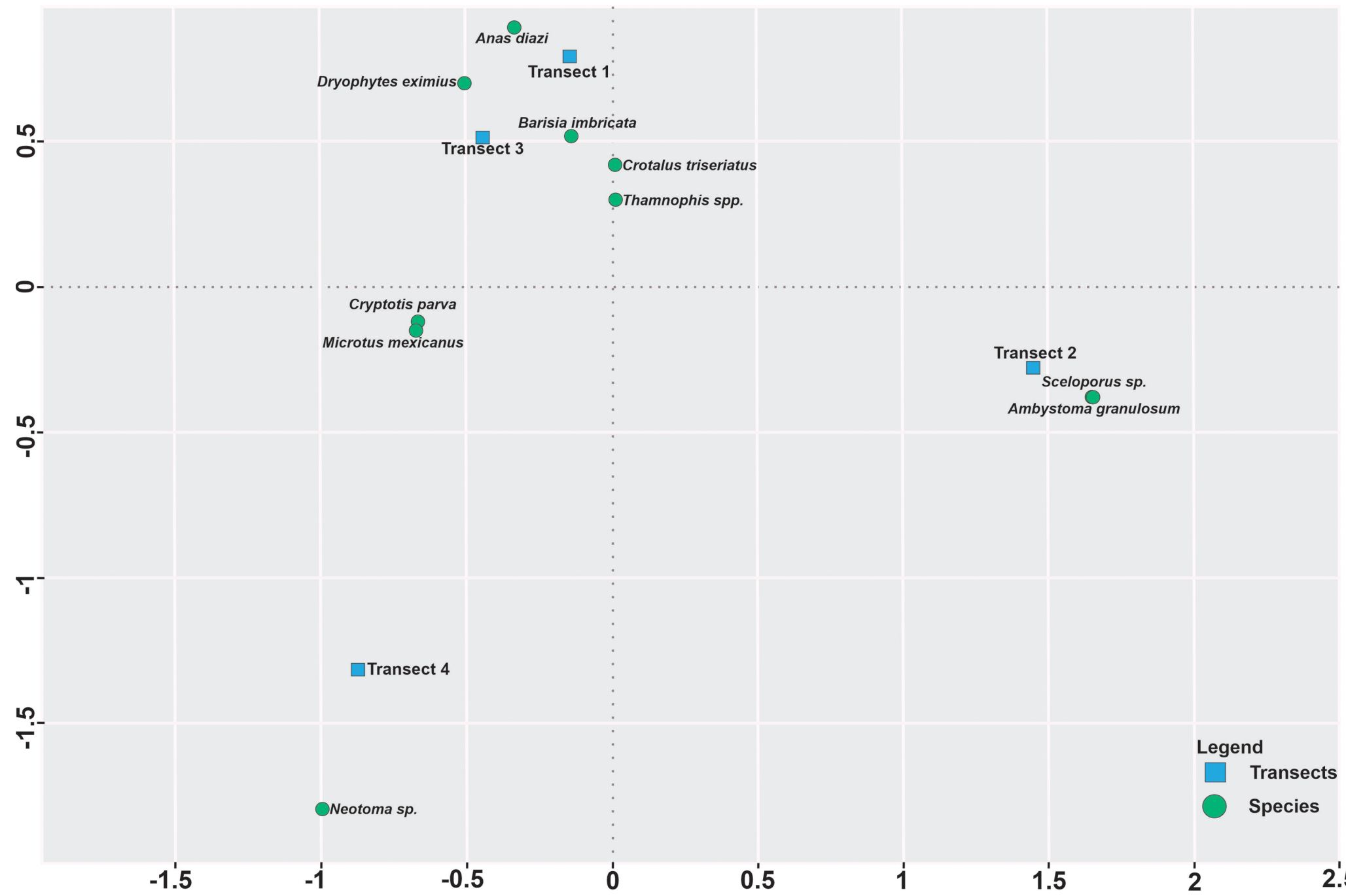

**Figure 5 Spatial correspondence analysis between species and transects along the open-channel irrigation system in "Area for the conservation and ecological research, Bordo Las Maravillas", State of Mexico, Mexico.** Green dots represent the species and blue squares represent the transects.

*sp*, and *A. granulosum* are associated with transect 2, while several species including *C. triseriatus, Thamnophis spp. B. imbricata* and others are mainly found in transects 1 or 3 (Fig. 5). Transect 1 had the highest heterogeneity ($H = 1.43$) and the lowest species richness ($S = 3$) and mortality percentage (3.6% of all dead individuals recorded). This transect is surrounded by native vegetation and is located within the protected area, it is not affected by the agricultural open-channel irrigation system and may be useful as a reference of the mortality impact of the irrigation system. Transect 2 ($H = 0.44$; $S = 5$; 14.39% of mortality) and 4 ($H = 2.99\text{E}{-5}$; $S = 5$; 22.30% of mortality) had the lowest landscape diversity; they presented an intermediate species richness and mortality. The landscape context in these transects is dominated by cropland. In contrast, transect 3 had intermediate landscape diversity ($H = 0.74$), but had the highest species richness ($S = 7$) and the highest percentage of mortality (59.71%). Again, the effect of carcasses removal or predation by some species must be taken in to account. Transects 2 to 4 may be more susceptible to this effect in comparison to transect 1.

# DISCUSSION

## Mortality

The studied irrigation channel system is directly affecting several small vertebrate species, mainly mammals and reptiles. Moreover, we may be underestimating vertebrates mortality in this system. If we consider that some carcasses may be removed by predators or

scavengers (*Barrientos et al., 2018*; *Prosser, Nattrass & Prosser, 2008*), and that live animals that we rescued from the channel and the floodgates would likely die, the mortality rate in this study could increase dramatically. It is possible that for some species the open-channel irrigation system does not represent a population threat, species as *M. mexicanus* or *B. imbricata* may be scarcely affected due to its relatively high reproductive rate, which may allow the species to maintain stable populations by replacing dead individuals, whilst some other species as *C. parva* or *Thamnophis spp.* may be more affected (Fig. 2). In order to clearly determinate this impact, more studies are needed where the mortality caused by the irrigation channel can be related to local population dynamics of each species and then determine if the irrigation channel may put the species persistence at risk. Nonetheless, our study does show that the open-irrigation channel is functioning as a pitfall trap that affects small species.

Our results show that mortality of small vertebrates varies according to three main factors, namely: structures of the open-channel irrigation system (*i.e.,* concrete channel and floodgates; Fig. 3), seasonality (*i.e.,* wet, and dry; Fig. 4) and landscape heterogeneity (*i.e.*, high, medium, and low landscape diversity; Fig. 5). Most of the affected species (*e.g.*, *B. imbricata, C. triseriatus*, *Thamnophis spp*.) seem to get into the channel but cannot climb out and die. Some other species, including small mammals such as *M. mexicanus* can also enter easily, but apparently move along the channel until they fall into the floodgates and drown. Mortality is significantly higher in the rainy season, being *C. triseriatus* and *M. mexicanus* the most affected species (Fig. 4). It has been suggested that the mortality peaks are related to the reproductive or dispersal periods of the species and the peaks of lower activity to inactivity in winter (*García, 2009*). Although we work on a relatively small area, our results on landscape diversity show that the transects are associated with different landscape contexts that include native vegetation, crops, and a mixture of both (Fig. 5). In our study area, landscape heterogeneity may be also influencing mortality rate. It has been showed that heterogeneous landscapes generally harbor more species and highest abundances than the homogeneous ones (*Arroyo-Rodríguez et al., 2019*). This shows that even in small and anthropized areas, as the one we studied, habitat heterogeneity may play an important role in species richness and should be managed to favor its conservation.

Wildlife mortality in linear infrastructures is frequently estimated through carcasses counting; nonetheless, two main factors must be considered, the overlooking of carcasses (*i.e.*, the probability of a researcher not finding a carcass in the field), and the probability of carcass disappearing due to removal by predators, scavengers, or other means (*Barrientos et al., 2018*; *Korner-Nievergelt et al., 2015*; *Prosser, Nattrass & Prosser, 2008*). Based on the characteristics of the studied infrastructure (its dimensions, and the absence of elements that may prevent carcasses detection), we considered that the probability of overlooking carcasses during our study was very low. Otherwise, the probability of carcasses removal may be considerable. It has been showed that body mass is negatively related to carcass persistence. Thus considering the small size of the species we are reporting, it is very likely that our estimates of mortality are underestimating the effect of the open channel irrigation system on small vertebrates.

To our knowledge, there is no studies that could help us determine which species may be acting as predators or scavengers in the study area. Nonetheless, the presence of feral dogs (evidenced by scats and footprints) seems to be very high. Feral cats may be also present in the area, although its presence is not as evident as the dogs. Several studies have evidenced the great impact of feral dogs and cats on wild species, especially in human populated areas (*Coronel-Arellano et al., 2021*; *Guedes et al., 2021*; *Ramos-Rendón et al., 2023*; *Young et al., 2011*). We think these species are the main predators and scavengers affecting our carcasses detection. Other wild species include the ferret (*Neogale frenata*), and the cacomixtle (*Bassariscus astutus*); both are also common, *B. astutus* is frequently reported in urban settlements whilst *N. frenata* is more common in rural areas. Otherwise, considering the characteristics of the irrigation channel, it seems more probable that predators and scavengers prey mainly on individuals or carcasses located on the concrete channel, as water level in the floodgates may prevent depredation or carcasses removal (water level is normally below the concrete channel, forming a pitfall trap). If we consider the irrigation channel structures where the species where detected, it is probable that mammals and amphibians' detection are less affected by predators and scavengers than lizards and snakes' detection.

Our study area can be considered a representative landscape in the region (more than 1,500 $km^2$), as similar systems (crop farms with adjacent water bodies), are used throughout as the functional unit for agricultural activities. Variables related to vegetation, such as cover area, height, and species composition, have been shown to be important in explaining the presence and abundance of species in irrigation channels; therefore, management practices for vegetation in channels are important for biodiversity conservation (*López-Polomares, López-Iborra & Martín-Cantarino, 2015*). Our results also suggest that landscape differences are associated with small vertebrate occurrence and mortality, even at this fine spatial scale (Fig. 5). Landscape contexts similar to those we studied (*i.e.,* croplands associated with small water bodies) are widespread through the world, and thus, may be exerting significant negative impact on small vertebrates. *Malano, Chien & Turral (1999)* reported that more than 275 million hectares are irrigated worldwide and that the irrigated area is steadily increasing by 1.5% per year. Considering this trend, we argue that, first, the impacts of irrigation systems on small vertebrates may be underestimated and, second, this issue deserves much more attention than it has received.

According to the IUCN red list of threatened species, the species affected by the irrigation channel are considered Least Concern (LC), except for the granular salamander (*A. granulosum*), which is considered Endangered (EN; *IUCN, 2020*). However, according to Mexican standards NOM-059-SEMARNAT-2010 (*SEMARNAT, 2010*), the small-eared shrew (*C. parva*) and two endemic species, the granular salamander (*A. granulosusm*) and the transvolcanic alligator lizard (*B. imbricata*), are under special protection (PR). On the other hand, two endemic species (*T. scalaris* and *T. scaliger*) as well as *T. eques* and the Mexican duck (*A. diazi*) are species in the threatened category (A). Unfortunately, there is no habitat management or any other strategy for conserving these vertebrates in the study area. Our results evidence that the irrigation channel is causing non-natural mortality of these species, and thus, it is necessary to take action in this issue. The affected species are

small predators that feed mainly on invertebrates and small vertebrates (*Álvarez-Castañeda & Reid, 2016*; *Manjarrez, García & Drummond, 2013*); and some of them are threatened in parts of their range by the conversion of forest to agricultural land. They act as controllers of pest populations, making it necessary to adopt protective measures to ensure the provision of these ecosystem services, especially on agricultural land (*Tuberville et al., 2005*).

Our results indicate that, although its relatively small size, the open-channel irrigation system studied is a threat to small vertebrate populations. Conservation efforts should be directed towards the protection of biodiversity by restructuring the open-channel agricultural irrigation system. The habitat loss at the patch and landscape level poses the greatest threat to all groups of vertebrates and this effect is amplified for species with relatively low dispersal such as small terrestrial micromammals, amphibians, and reptiles. Therefore, the conservation strategies require the implementation of initiatives focused on preventing further habitat degradation (*Dickman & Doncaster, 1987*; *Ramos-Lara & Gómez-Ortiz, 2019*; *San-José, Arroyo-Rodríguez & Sánchez-Cordero, 2014*). Management should focus on maintaining micro connectivity between vegetation patches, depending on the characteristics of the affected species, different options may be applied to reduce their mortality. It is suggested that vegetation management be improved in collaboration with growers and owners, so that the presence of vegetation within the channel be allowed, which acts as a cost-effective and immediate measure to allow escaping of individuals that have fallen into the channel.

Some research has examined effectiveness of different alternatives applied in irrigation channels to prevent wildlife mortality (*Albanesi, Jayat & Brown, 2016*; *Baechli, Albanesi & Bellis, 2021*; *Bucci & Krausman, 2015*; *Gačić, Danilović & Đorđev, 2013*; *García, 2009*; *Peris & Morales, 2004*; *Rautenstrauch & Krausman, 1989*). These alternatives include those that restrict access (*e.g.*, permanent fences, and grids), provide landscape continuity (*e.g.*, overpasses, bridges and wildlife crossings), facilitate escape (*e.g.*, wooden or concrete stairs for wildlife escape, ramps without slippery surfaces, and dunes), and those specifically designed to prevent animals from using the channels in search of resources (*e.g.*, water catchments; (*Albanesi, Jayat & Brown, 2016*; *Bucci & Krausman, 2015*; *Gačić, Danilović & Đorđev, 2013*). These mitigation alternatives should be in areas with high vegetation cover or near water bodies, as this increases the likelihood of wildlife use, as is the case with railway lines (*Clair et al., 2020*).

## Funding

This work has been supported by field materials from the IdeaWild project. The funders had no role in study design, data collection and analysis, decision to publish, or preparation of the manuscript.

## Grant Disclosures

The following grant information was disclosed by the authors:
IdeaWild project.

## Competing Interests

Armando Sunny is an Academic Editor for PeerJ.

## Author Contributions

- Yuriana Gómez-Ortiz conceived and designed the experiments, performed the experiments, analyzed the data, prepared figures and/or tables, authored or reviewed drafts of the article, and approved the final draft.
- Hublester Domínguez-Vega conceived and designed the experiments, performed the experiments, analyzed the data, prepared figures and/or tables, authored or reviewed drafts of the article, and approved the final draft.
- Leroy Soria-Díaz conceived and designed the experiments, performed the experiments, authored or reviewed drafts of the article, and approved the final draft.
- Tamara Rubio-Blanco performed the experiments, authored or reviewed drafts of the article, and approved the final draft.
- Claudia C. Astudillo-Sánchez performed the experiments, authored or reviewed drafts of the article, and approved the final draft.
- Victor Mundo conceived and designed the experiments, performed the experiments, authored or reviewed drafts of the article, and approved the final draft.
- Armando Sunny performed the experiments, analyzed the data, prepared figures and/or tables, authored or reviewed drafts of the article, and approved the final draft.

## Animal Ethics

The following information was supplied relating to ethical approvals (*i.e.*, approving body and any reference numbers):

The Secretaría del Medio Ambiente y Recursos Naturales (SEMARNAT), provided permissions for this work through the number (FAUT-0351).

## Data Availability

The raw data is available in the Supplementary File.

## Supplemental Information

Supplemental information for this article can be found online at http://dx.doi.org/10.7717/peerj.17818#supplemental-information.

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
