# Peer review of "Negative effects of agricultural open-channel irrigation system on vertebrate populations in central Mexico"

_PeerJ, doi:10.7717/peerj.17818_

## Round 0.1 · original submission · Major Revisions

Dear Dr. Hublester Domínguez-Vega,

The manuscript entitled “Negative effects of agricultural open-channel irrigation system on vertebrate populations in central Mexico” was reviewed by two independent reviewers. Both reviewers were positive about the manuscript but also raised significant concerns with the current presentation of the manuscript. Reviewer#1 found the information in the introduction insufficient and suggested some clarification along the text. Reviewer#2 was concerned about the lack of reporting in the methods and results sections and potentially inadequate statistical analyses. I agree that the manuscript needs significant changes. I hope the comments from the reviewers are helpful.

·

Basic reporting

I think that the data obtained here was valuable.
On the other hand, the information in the introduction is insufficient. The significance and objectives of this study need to be clarified through a more detailed review of the relationship with previous studies.
In line 74, I think it's better to add "modern" before "agriculture" because traditional agriculture was thought to provide compensatory vegetation in wetlands in Asia (ex Primac and Kobori 2003).
In line 75, there is mistype (IUCN).
I think you need to clarify the scope of this study (such as what you wrote in line 221 to line 232) before the declaration of the objective in line 89. I think focusing on small vertebrates and small channels (or ditches) became a good reason to conduct this study in this area.

Experimental design

In the carcasses research, I think it has to be thought about removal rates by predators and scavengers. The removal rate went up to 76% in 24h in the summer in the UK (Prosser et al. 2008), therefore there must be some bias in the carcasses left behind. Scavenger prey preference and distribution may also affected. Please add some information about the removal rate in this area.
I could not understand the necessity of the part of reviewing conservation and management strategies because Table 1 only appears here with no discussion and it is from Lemly et al 2000.

Validity of the findings

As I wrote in basic reporting, the paragraph from line 221 to line 232 should be written in the "introduction". Overall, the discussion is too long and too much generalized. Repetition, such as several transects in similar environments, is necessary to generalize the results.
The discussion should be based on the results obtained. Neither figures nor tables are cited and it is unclear which part of the discussion is derived from the results.

·

Basic reporting

The paper has a good start and most of the pertinent information is reported, but multiple sections need additional reporting. The introduction and discussion contain sufficient information and citations, but minor edits are needed. The figures/table are adequate. The authors may need to provide additional raw data. See my in-depth comments.

Experimental design

Overall, the experimental design seems sufficient. The authors do a good job of describing the knowledge gap and the need for more studies on this topic. However, the authors need to re-assess their statistical analyses, and further describe their methods and results. See my in-depth comments.

Validity of the findings

The authors need to re-assess their statistical analyses, and further describe their methods and results. See my in-depth comments.

Additional comments

Overview - Mortality of wildlife from anthropogenic causes is a very important issue and I applaud the authors for their work on this topic. I would love to see this paper published after a few key issues are addressed. The main issue with the paper is the lack of reporting in the methods and results sections and potentially inadequate statistical analyses. See my in depth comments on this below. The next issue is the literature review section. This section needs to be either removed or greatly edited and improved. Multiple topics throughout the paper need to be further explained. There are minor grammar edits throughout. Overall, the paper has a fantastic start and contains valuable information!

Grammar edits:
There are other minor grammar edits throughout the paper such as misuse of commas.
A common grammatical error throughout the paper is incorrect use of semicolons. The proper use of a semicolon is: “used to indicate a major division in a sentence where a more distinct separation is felt between clauses or items on a list than is indicated by a comma, as between the two clauses of a compound sentence.” The authors use semicolons to connect clauses that should actually be separate sentences. Below, I point out many of these instances. However, the authors should thoroughly check each sentence that contains a semicolon for proper use.
Another grammatical edit is being careful about using “they, their, its, etc.” to start a sentence following the sentence where the word was described. For example, the author states “The impact of irrigation channels (and also other linear infrastructure), includes two main types of disturbance; the most visible, is the direct impact through mortality of individuals. Nevertheless, THEY also indirectly affect wildlife…” The word “they” is used to describe the channels that were discussed in the previous sentence. I would recommend replacing “they” with something more descriptive, such as irrigation channels, so that the sentence is more clear. Using “they” to describe something that was stated in the previous sentence can be confusing for the reader. It is better to be explicit. I recommend that the author checks for this scenario throughout the paper and edit if needed.

Section by section comments:
Abstract:
The abstract is overall very good, but needs some grammatical edits.
Introduction:
The authors have a good start on the introduction. However, I think it needs to further explain a couple of topics such as what other research has been conducted on this topic. What have other studies found? Be more descriptive. Also, more citations are needed in areas where addition topics are brought up. For example, the authors discuss other causes of wildlife mortality such as roads, railways, power lines, etc. All of these topics need citations and may need to be discussed in further detail. Lastly, the objectives need to be assessed again. The author states only two objectives, but it seems as though more objectives were met than those mentioned. Be more descriptive.
Methods:
The methods section is missing key information on survey methodology such as when the surveys conducted, how were they conducted (on foot?), by how many biologists, using what equipment, etc. These are critical pieces of information that need to be in the paper. The “Statistical analysis” section needs a lot more detail and description. There needs to be clear details on how each test was conducted, what tests were conducted, how were they conducted/set up, what software was used, etc. This section is vague and it makes the results hard to understand.
Methods section “Conservation and management strategies” – This section needs to be assessed. If a literature review was indeed conducted, there needs to be far more information in this section. What were your methods/criteria for finding papers? How many papers were assessed? Then, there needs to be far more information in the results section on what was found during the literature review. This usually involves additional statistical analyses or reporting of what was found in the papers assessed. If a thorough literature review was not conducted, this section needs to be removed from here and the objectives statement.
Results:
Overall, there is a huge lack of detail and description in the results section. Far more information needs to be reported in order for the reader to understand how the tests were conducted and how to interpret them. There may be issues with inadequate tests and pseudoreplication. I recommend the authors look into Hurlbert 1884 “Pseudoreplication and the Design of Ecological Field Experiments” to ensure there are no issues with pseudoreplication here.
Additionally, the use of percentages throughout the results section can be confusing. Percentages are easily misunderstood, so make sure that there is explanation behind each percentage being reported.
Results section “Occurrence and mortality between channel and floodgates” – I’m not sure that a chi-square test is appropriate here. It seems as though you could simply state the counts of dead animals in floodgates vs channels. It is not clear what exact variables were included in the test and how they were compared. The design and results of this test needs to be re-assessed and further explained.
Results section “Mortality differences between dry and wet seasons” – Same issues as above section. Re-assess chi-square analysis.
Results section “Mortality differences between habitat features” – Far more information is needed here to explain how the land use diversity index was designed and used.
Results section – “Conservation and management strategies” – See the above comments about this section.
Discussion:
Overall, this section is well-written. See my line-by-line comments below for edits in this section.
Figure captions:
Figure captions should be descriptive enough to stand alone. In other words, they should be descriptive enough for the reader to understand the context by just reading the caption. These captions need to add some more information such as dates of the study and methods (during transect surveys of channels, etc.). Apply this to all of the figure captions.
Raw data:
These data are already processed/analyzed data. My understanding is that the data provided should be in a more raw form with attributes such as date of survey, species, biologist, transect, etc.

Line-by-line edits:
Line 24 – delete comma in between “channels” and “produce”
Line 26 – delete semicolon and start new sentence at “Nonetheless”
Line 27 – add on to the sentence ending in “little attention.” Received little attention by who? Perhaps by conservation biologists.
Line 28 – delete “the” in between “to” and “different”
Line 30 - delete "the” at the start of the line
Line 30 – should add the length and location of the transects
Line 31 – “We examine” should be “We examined”
Line 32 – delete comma in between “analysis” and “and”
Line 34 – The word “Besides” doesn’t quite work here. A better word could be “Additionally,” or “Further,”
Line 36 – add “the” in between “analyze” and “conservation”
Line 63 – citations should be added to the end of the sentence on line 64
Line 66 – could add “to vertebrates” or “wildlife” after “of disturbance” to be more descriptive
Line 66 – delete comma in between “visible” and “is”
Line 67 – The word “Nevertheless” doesn’t quite work to start the sentence with. I recommend deleting it. However, if doing so, I recommend starting the sentence with something more descriptive than “they.” See my comments on the use of “they”.
Line 71 – needs citations at the end of the sentence ending in “locally extinct”
Lines 65-73 – This paragraph should be expanded and reworked. The author briefly discusses three different topics: disturbance via irrigation channels, linear infrastructure mortality, and other studies on mortality and mitigation measures. Each of these should be further described, and perhaps separated into separate paragraphs.
Line 77 – Regarding “their daily foraging” – a better word may be “movement” because wildlife move around the landscape for other purposes than foraging (dispersing, socializing, etc.) that could cause them to fall into a channel.
Line 77 – delete/fix semicolon
Line 78 – Regarding “depending on their dimensions, these structures can affect vertebrates of all sizes” – this sentence should be further explained.
Line 85 – Replace “Their” to something more descriptive.
Line 89 – Replace “is” to “was”
Line 89 – delete “the” in between “to” and “structures”
Lines 88 – 93 – The objectives statements should be expanded. There were more objectives in this paper than the two stated. For example, the authors also assessed mortality in dry vs. wet seasons. This should be stated here along with any other missing objectives.
Line 101 – add citation for Figure 1 at the end of the sentence
Lines 102-112 – This is a very good paragraph that describes the study area well. Well done here. This amount of description should be used throughout the paper.
Line 109 – this is a strange place to site Figure 1. I recommend changing it to Line 101 (see above)
Line 114 – Regarding “approximately two years” – use exact dates here!
Line 116 – The methods section is missing information on survey methodology such as how often the surveys were conducted, how were they conducted (on foot?), by how many biologists, using what equipment. These are critical pieces of information that need to be in the paper. In between the two sentences in this line is a good place to put this information.
Line 117 – delete semicolon and start a new sentence at “All”
Lines 119-121 – This statement needs to go somewhere else. Maybe at the end of the section “Surveys of wildlife and habitat features”
Line 142 – change “investigate” to “investigated”
Line 144 – Regarding sentence ending in “more commonly found” – further describe. More commonly found in relation to what?
Lines 147 – 151 – This section needs to be expanded. What were your methods/criteria for finding papers? How many papers were assessed? Much more information is needed here. See my comments above about this section and if it’s needed or should be deleted.
Line 154 – colon is not needed
Lines 153-159 – should add how information on how many live animals were found
Lines 162-163 – This sentence should be reworded. What do you mean by “these”? Significantly different between which variables?
Line 184 – delete semicolon and “this is” and start next sentence with “Some”
Line 188 – recommend deleting or replacing the word “clearly”
Line 189 – “a pack of” – is not a great term for this. I recommend deleting these words. “…while several species…” works just fine
Line 196 – delete semicolon
Line 200 – delete comma after “system” and start a new sentence at “Although”
Line 202 – I recommend starting the sentence with something other than “This”
Lines 206-109 – This information is incredibly important! Great start! But there should be way more information, results, and citations here about what other papers found. Expand this section!
Line 211 – replace “greater” with “greatest”
Line 213 – delete semicolon and start new sentence
Lines 213-214 – Further explain what management strategies could help avoid drowning of voles and reptiles. Be descriptive!
Line 215 – capitalize “table”
Line 233 – regarding the first sentence – adjust to be more descriptive of your results. “may be high” does not adequately describe your results.
Line 234 – “was recorded only by dead individuals” – this phrase does not make sense. Adjust to be more descriptive.
Lines 234-237 – These sentences need to be reworded. There are confusing.
Line 240 – “be on this situation” need to be reworded
Line 240 – regarding “high reproductive rate” – reproductive rate has not been discussed anywhere earlier in this paper. Either further explain here, or add some information to the introduction about how reproduction rate may influence mortality.
Line 241 – “May” shouldn’t be capitalized
Line 241 – fix semicolon
Line 242-243 – regarding “local population size” – again, population size has not been discussed in this paper. Either delete or expand.
Lines 233-244 – Overall, this paragraph needs work. It is confusing and contains some statements that I’m not sure belong in the paper. Please re-assess this paragraph.
Line 249 – change “this” to something more descriptive. I’m not sure what the author is talking about when they say “this.”
Line 256 – regarding “we think” – I recommend changing these words. Make statements based on data. Don’t speculate.
Line 277 – The sentence shouldn’t start with “These.”
Line 278 – replace “underestimate” to “underestimated”
Lines 280-291 – There is valuable information here! I think this paragraph could be expanded. Expand on what species’ status means in terms of mitigation. What are their protections?
Lines 293-302 – This paragraph explains some of the most valuable information of this paper! Great job here.
Line 307 – change “has” to “have”
Line 307 – delete comma in between “used” and “so”
Line 308 – fix semicolon
Line 309 - not sure what the author means by “built”
Line 314 – fix semicolon
Line 314 – this sentence is slightly confusing and should be further explained
Line 320 – needs citations at end of sentence
Line 325 – regarding “low dispersal” – I’m not sure if this should be included here. The authors did not test this
Lines 328-329 – delete “In the studied landscape” – it is not needed
Line 452 – should add the number of papers assessed in this literature review. See other comments on the literature review.
Figure 1 caption – fix semicolon
Figure 2 caption – add “found” or “detected” in between “species” and “in”
Figure 5 caption – further explain this figure. What do the axes mean, etc. Also, delete “the” in between “and” and “transects”

---

## Round 0.2 · Minor Revisions

The revised manuscript was reviewed by the original reviewers and both were positive about authors's replies and the revised document. However, Reviewer#1 still suggested some minor changes. I hope you find reviewers' comments useful.

·

Basic reporting

I think that “for what” should be written before the colon on line 89.It is not clear what you want to know and why you are conducting the three analyses. I think you need a sentence stating the reason, e.g., “to understand whether there is a direct impact of the irrgation cannal on small vertebrates and the conditions that make a difference in the impact."
In line 220, I don't understand what this sentence mean. Are "small mammals and reptlees" same as "several vertebrates species ”? Or you mean some mammals or reptiles eat or attack the other vertebrates in the cannal ? If so, you should write as is.

Experimental design

No comment

Validity of the findings

Please make sure the limitation of what you can mention with your results.
I think you need to discuss more about effect of predators and scavengers such as what kind of predators and scavengers are living around the study area and which kinds of animals are thought to be under estimate. It’s also necessary to mention about the effects of those scavengers at each analysis.

·

Basic reporting

The authors have updated the paper in response to mine and one other reviewer's comments. The "Basic reporting" of this paper is of high standard.

Experimental design

The authors have updated the paper in response to mine and one other reviewer's comments. The experimental design in this paper is sound and well explained.

Validity of the findings

The authors have updated the paper in response to mine and one other reviewer's comments. Their findings are of high impact and explained well.

Additional comments

The authors did a phenomenal job at integrating mine and the reviewer's comments. This is a great paper and I'm excited to see it published.

---

## Round 0.3 · accepted · Accept

I have checked authors's replies, previous reviewers's comments and the revised manuscript and I am happy to see that authors addressed all suggestions received. I acknowledge authors for taking the time and effort to revise and improve the manuscript.